# Lesbian and Gay Population, Work Experience, and Well-Being: A Ten-Year Systematic Review

**DOI:** 10.3390/ijerph21101355

**Published:** 2024-10-14

**Authors:** Marina Lacatena, Ferdinando Ramaglia, Federica Vallone, Maria Clelia Zurlo, Massimiliano Sommantico

**Affiliations:** 1Department of Humanities, University of Naples Federico II, Via Porta di Massa 1, 80133 Naples, Italy; marina.lacatena@unina.it (M.L.); federica.vallone@unina.it (F.V.); 2Department of Research and Humanistic Innovation, University of Bari Aldo Moro, Piazza Umberto I, 70121 Bari, Italy; ferdinando.ramaglia@uniba.it; 3Department of Political Sciences, University of Naples Federico II, Via Porta di Massa 1, 80138 Naples, Italy; zurlo@unina.it

**Keywords:** LG workers, intersectionality, heterosexism and heteronormativity, outness and disclosure, organizational climate

## Abstract

Despite an increase in the promotion of equal opportunities at work, there is still persistent discrimination against lesbian and gay (LG) workers. In this vein, this study aimed to systematically review the research investigating the peculiarities of the work experience of LG people, particularly considering the theoretical frameworks in the approach to sexual minorities’ work-related issues, as well as individual and contextual variables influencing the work experience and the impact they may have on health and well-being. We explored the PsycArticles, EMBASE, Scopus, and Web of Science electronic databases and the EBSCOHost (PsycInfo, Psychology and Behavioral Sciences Collection) scholarly search engine, between 01/01/2013 to 01/03/2023, with regards to the search terms “lgb*”, “gay*”, “lesbian*”, “homosexual*”, and “sexual minorit*”, associated with “employee*”, ”personnel”, “worker*”, and “staff”, and with “workplace”, “work”, “job”, “occupation”, “employment”, and “career”. Data were narratively synthesized and critically discussed. Of the 1584 potentially eligible articles, 140 papers contributed to this systematic review. Five main theoretical frameworks were identified: (a) minority stress, (b) sexual prejudice and stigma, (c) queer and Foucauldian paradigms, (d) social identity theories, and (e) intersectionality. Furthermore, significant individual (e.g., outness, disclosure, and work–family conflict) and contextual (e.g., heterosexist and heteronormative workplace climate and culture) variables influencing LG people’s work experience were identified. This review highlights the need to develop a unified theoretical model for the construction of specific measurement tools to assess the work experience of LG people and for the implementation of interventions aimed at minimizing the effects of stigma in work contexts.

## 1. Introduction

In recent decades, the debate on lesbian, gay, bisexual, transgender, and queer (LGBTQ+) issues has been the focus of political and media attention, both in its more positive declinations of openness, recognition, and deconstruction of old discriminatory myths and habits and in its more negative aspects, such as the persistence and exacerbation of discriminatory forms by individuals, communities, or states towards the expansion of this liberalizing push for sexual orientations and gender identities. In this polarized social climate [1], research on LGBTQ+ community issues has become increasingly necessary to produce up-to-date and original knowledge that deals with analyzing and deconstructing negative behaviors, beliefs, and affectivities related to homo–bi–trans–queerphobia and the social stigma surrounding gender identity and sexual orientation, and such is aimed at devising interventions useful for implementing psychophysical well-being.

Chapter 3 of the Charter of Fundamental Rights of the European Union (EU) [2] enshrines people’s equality and the right to non-discrimination. The European Commission establishes schemes and laws to promote equality for LGBTQ+ people in Europe, which suffers from wide territorial heterogeneity, as evidenced by the ILGA 2018 Europe section’s laws and policies [3].

For example, on 24 March 2022, an ISTAT (Italian National Institute of Statistic) and UNAR (National Office Against Racial Discrimination) survey involved approximately twenty-one thousand LGBTQ+ people—in current or past civil unions and residing in Italy as of 1 January 2020—with the aim of exploring employment discrimination against LGBTQ+ people [4]. Although the results cannot be considered representative of the LGBTQ+ Italian population, some of these data could be relevant to this research. Indeed, 26% of employed or formerly employed people reported that being homosexual or bisexual has been a disadvantage in terms of their working life, career, and professional growth, recognition or appreciation of their professionalism, and regarding income and pay. Concerning their last job, 40.3% avoided talking about their private lives and associating with colleagues in their free time to keep their sexual orientation hidden and to reduce the risk of revealing it. Furthermore, about six out of ten people have experienced at least one form of micro-aggression in the workplace related to their sexual orientation, with micro-aggression being defined as sending disparaging messages and subtle insults to minority individuals within short daily exchanges, often of an unconscious nature [5,6,7]. A high percentage (34.5%) of employees have experienced at least one discrimination event during their employment, with a higher incidence among women and bisexual individuals and a predominance in individuals employed on fixed-term contracts, thus suggesting a more protective condition for those working in public settings. Finally, about one in five experience a hostile climate or aggression in their work environment, such as slander, mockery, and verbal humiliation. Therefore, despite an increase in the promotion of equal opportunities at work, both subtle and blatant discrimination still exist.

Discrimination against LGBTQ+ people at work is, as stated by Anastas [8], “a form of violence that denies them full participation in essential social and economic activities and institutions, perpetuates economic injustice, and reduces their opportunities to realize human potential” (p. 84). It is also a direct violation of Article 23 of the 1948 United Nations Universal Declaration of Human Rights [9] on the universal right to work, free choice of employment, just and favorable work conditions, protection against unemployment, and equal pay for equal work [10].

In occupational research, a tailored focus has been increasingly given to developing research and interventions to prevent stress and discrimination in the workplace. This is also due to the growing recognition of the doubtless evidence that promoting workers’ psychological and relational health conditions will benefit the quality of the work organization and society as a whole [11,12].

For this perspective, according to one of the main theoretical frameworks for evaluating occupational well-being, namely the job demand–resources model [13], there is always a kind of interplay of forces within work contexts. Indeed, the latter model [13] allows researchers to simultaneously investigate the effects of the interplay between perceived job demands (physical, social, or organizational aspects of the job requiring physical and/or mental effort and associated with physiological and psychological costs) and perceived job resources (physical, social, or organizational aspects of the job that can be instrumental in achieving work goals, reducing job demands, and/or stimulating personal development) in terms of workers’ wellbeing. Specifically, if workers perceive high demands along with low resources (i.e., imbalance between perceived demands and resources), this state of fatigue (e.g., due to workplace conflicts) can have adverse outcomes on workers’ health, especially in terms of anxiety and depression [14].

However, the work experiences of LG people may require tailored research attention, also given the higher risk they are exposed to in terms of further specific sources of stress and discrimination. Indeed, “minority stressors have a unique negative effect on health and well-being that cannot be reduced to stress in general” [15] (p. 2), and experiences of stigma and rejection are correlated with significantly higher rates of depression and anxiety than in the heterosexual population [16,17,18,19,20].

Therefore, this systematic review aims to (1) identify the theoretical frameworks used in the approach to sexual minorities’ work-related issues and (2) identify the main individual and contextual variables influencing the work experience of LG people.

## 2. Methods

### 2.1. Database Search

This review was conducted between March 2023 and June 2023, following the Preferred Reporting Items for Systematic Reviews and Meta-Analyses (PRISMA) guidance [21] (See PRISMA 2020 checklist in the Appendix A). Furthermore, as a review of preexisting study reports, the Psychological Research Ethics Committee of the Department of Humanities of the University of Naples deemed it exempt from ethical approval. The study was registered on the Open Science Framework (OSF) platform (https://doi.org/10.17605/OSF.IO/5DH92, accessed on 29 May 2023).

We conducted an electronic search of four electronic databases and one scholarly search engine between 1 January 2013 and 1 March 2023, namely EBSCOHost (PsycInfo, Psychology and Behavioral Sciences Collection), PsycArticles, EMBASE, Scopus, and the Web of Science. There were no language restrictions. The following keywords were used in a [Abstract] search: “(lgb* OR gay* OR lesbian* OR homosexual* OR sexual minorit*) AND (employee* OR personnel OR worker* OR staff) AND (workplace OR work OR job OR occupation OR employment OR career)”.

The reference lists of the identified studies were searched to find relevant articles and to ensure that all related publications were included in the analysis. The full-text versions of the literature were screened and analyzed for methodologic quality. Three research team members performed the processes independently, and disagreements were resolved by discussion and consensus with a fourth member of the research team.

### 2.2. Selection: Inclusion and Exclusion Criteria

We screened all observational studies analyzing LG workers’ experience. The specific inclusion criteria were as follows: (a) publication date between 1 January 2013–1 March 2023; (b) original research articles published in all languages; and (c) studies reporting qualitative and/or quantitative data. Specific exclusion criteria were as follows: (a) abstracts; (b) letters; (c) editorials; and (d) commentaries (see Figure 1).

Furthermore, a quality assessment was conducted for all the articles that met the inclusion criteria. The Mixed-Methods Appraisal Tool (MMAT) version 2018 [22] was used to assign the quality rating. The MMAT is a critical appraisal tool widely used in systematic mixed studies reviews since it allows the appraisal of the methodological quality of the main categories of studies, namely qualitative research, randomized controlled trials, non-randomized studies, quantitative descriptive studies, and mixed methods studies. Three reviewers (M.L., F.R., and F.V.) were firstly involved in the appraisal process and independently assigned the quality rating to each paper according to the study design category (five specific methodological quality criteria for each category; rating 0–2; range: 0–10). Any discrepancy or disagreement was solved by discussions supervised by M.S. and involving all the authors. Studies reporting a score ≥ 5 were included in the final analysis.

The data extraction and synthesis were performed narratively considering the objectives settled for the purposes of the current review study, namely (1) to identify the theoretical frameworks used in the approach to sexual minorities’ work-related issues and (2) to identify the main individual and contextual variables influencing the work experience of LG people.

## 3. Results

### 3.1. Characteristics of Selected Literature

A total of 1548 records were identified through an electronic search. When duplicates were removed, 987 records remained and were screened. Of these, 744 records were excluded (as being not relevant, abstracts, letters, and editorials). A total of 243 potential papers were assessed for eligibility. Of these, 103 papers were excluded (commentaries and reviews). One hundred forty papers were judged relevant and contributed to this systematic review (see Appendix A). All the papers were evaluated as reaching a quality score of ≥5 in MMAT [22].

The 140 studies considered involved more than 14.116.688 million participants. Only one study (0.71%) [23] does not specify the sample size. Compared with the total number of participants, 344.974 identified themselves as homosexuals. Furthermore, 34 studies (24.29%) [16,23,24,25,26,27,28,29,30,31,32,33,34,35,36,37,38,39,40,41,42,43,44,45,46,47,48,49,50,51,52,53,54,55] used compound acronyms (LGB, LGBT, sexual minority, etc.) to name their populations, which does not allow precise estimation of the homosexual subsample.

Considering the study design, only nine studies (6.4%) [56,57,58,59,60,61,62,63,64] were longitudinal, while the majority used an observational/cross-sectional research design. Only 17 studies (12.1%) [26,33,47,48,52,56,65,66,67,68,69,70,71,72,73,74,75] were conducted with representative samples, while the majority were conducted with convenience, purposive, or theory-based samples. Furthermore, only 11 studies (7.9%) [76,77,78,79,80,81,82,83,84,85,86] compared populations from different countries.

Finally, sample descriptions vary significantly among studies. Indeed, only 36 studies (25,71%) [51,57,58,64,71,76,78,79,80,85,86,87,88,89,90,91,92,93,94,95,96,97,98,99,100,101,102,103,104,105,106,107,108,109,110,111] specify the population in terms of both sexual orientation and gender identity (see Limitations).

### 3.2. Theoretical Frameworks in the Approach to Sexual Minorities’ Work-Related Issues

The most represented theoretical frameworks in the approach to sexual minorities’ work-related issues—all clearly defined in the introduction section of the reviewed studies—are as follows: (a) minority stress theory [19] (16.4%, 21 out of 140 studies) [26,43,44,49,73,85,89,90,93,106,112,113,114,115,116,117,118,119,120,121,122] which has been used extensively for decades, allows for the multidimensional, unique, and complex experience of LG individuals in work contexts through the analysis of specific proximal and distal stressors that alter actual and perceived experiences; (b) sexual prejudice and stigma theories [123,124] (11.4%, 16 out of 140 studies) [32,33,75,84,94,110,117,120,125,126,127,128,129,130,131] focus on comprehending society’s negative regard for any behavior, identity, or community that is not heterosexual, the cultural ideology that perpetuates sexual stigma, and the negative attitudes based on sexual orientation with their consequences on LGBTQ+ individuals; (c) queer theory and Foucauldian paradigms [132,133] (10%, 14 out of 140 studies) [33,36,45,46,60,85,88,101,134,135,136,137,138] are largely present in contemporary policy debates on the topic and aim to deconstruct hegemonic stereotypes related to LGBTQ+ identities, posit the marginalization of sexual minorities as a given, and rethink concepts, such as resistance and non-assimilation; (d) social identity theories [139] (9.3%, 13 out of 140 studies) [34,40,51,59,77,78,86,95,99,130,140,141,142] refer to identity as a construct that continuously interacts with and is negotiated with work/organizational contexts and as a complex continuum, rather than a monolithic point. Furthermore, the concept of identity is considered most salient in a specific context that influences whether, when, and how to let one’s sexual identity emerge; and (e) intersectionality paradigm [143] (6.4%, 9 out of 140 studies) [34,38,43,63,81,92,96,117,144] turns out to be an essential key for a thorough understanding of the complexity of the self. In this sense, from the concept of identity, also understood as a social construction, it is possible to deduce that numerous variables can replay themselves in work contexts, such as ethnicity, gender, organizational hierarchical position, the type of work, and the specific place where it is performed.

Of the remaining 66 studies, 12 (8.6%) [16,52,55,57,58,62,69,125,145,146,147,148] present varied theoretical frameworks, such as: (a) the multilevel relational framework to diversity management [16], which proposes an interconnected and situated analysis of individual (at the micro-level, taking into account individual influences on equal opportunities), organizational (at the meso-level, assessing organizational approaches and strategies), and structural (at the macro-level, examining legal, institutional and socio-cultural structures) variables related to diversity management; (b) the communication theory of identity [145], which posits that people communicatively manage and construct identity with others through personal, enacted, relational, and group identity frames; (c) the social exchange theory [69], which posits that LG employee’s job satisfaction and affective commitment depend on the effect of their perceptions of organization’s inclusive work environment human resources practices; and (d) the disclosure process model [62], which posits that the decision to disclose one’s identity depends on the extent to which an identity management event activates approach goals (involving moving closer to a rewarding outcome, such as increased authenticity and relational intimacy) versus avoidance goals (involving moving away from a potential negative outcome, such as rejection and harassment).

On the other hand, 54 studies (38.6%) do not specify a clear theoretical framework of reference, essentially presenting generic reviews of the literature on the topic [29,48,80,87,99,149,150] and mainly focusing on heterosexism [64,104,151,152,153] heteronormativity [81,100,103,111,154], cisnormativity [27,41,67,82,105], and pressure to conform to masculine gender norms [56,155,156] in workplaces (see Table 1).

### 3.3. Individual Variables Influencing the Work Experience of LG People

Most of the studies address the issue of being or not being out in the workplace, as it is believed that this experience affects interpersonal relationships, performance, and well-being [25,77,108,157,158]. The first finding is that disclosure and outness can positively affect interpersonal relationships and social comfort [130]. Indeed, it has been found that, due to identity concealment, subjects may experience a decrease in positive affect and an increase in negative affect [62]. Outness, however, is not necessarily a precise moment but is represented as a continuum in individuals’ lives. In this vein, a recurring concept is that of visibility management, understood as “regulation of disclosure of one’s sexual orientation to maintain privacy and minimize stigma, harm or marginalization” [159] (p. 1). This is also a dimension strongly related to specific cultural and organizational contexts. In this vein, the concept of “negative face” [95] is interesting in exploring people’s desire to express their sexual identity freely, to emphasize both how life and work are in a mutually necessary relationship and how, in the workplace, identity is relational and socially produced. Therefore, the desire for authenticity turns out to be part of the process of identity negotiation [83,87,95,103].

A central issue, then, seems to be that of authenticity. For example, the State Authenticity as Fit to the Environment (SAFE) [160] model is effective in exploring how people with devalued social identities can feel comfortable in an “identity-safe context, according to which there are three types of adaptation that can lead to authenticity: self-concept fit, goal fit, and social fit” [57] (p. 3). It has emerged that disclosure is situated in different contexts, and indeed, many workers are out in the office but in the “closet” in the field, as well as that disclosure is predicted by perceived support from the potential recipient of this information [23,50,55]. Thus, negative aspects of authenticity also emerged, especially when implemented in contexts that do not provide psychological safety and do not support employees who choose to be out in the workplace [58]. But while it seems that the possibility of being openly oneself is a central issue for an individual’s subjective and relational well-being, it turns out to be equally true that being out does not always seem to be the right choice for personal and occupational safeguards [58,90]. Indeed, being out is crucial in the most challenging and most at-risk work contexts as a political strategy aimed at change [126]. In this vein, homonegative events can deter and encourage disclosure, understood as an act of resilience [103,104].

The review also highlighted different coping strategies implemented by LG people to manage challenges and difficulties in the workplace, such as finding safe spaces, negotiating identity, connectedness, having heterosexual allies, and having a context with company networks and policies that promote inclusiveness and safety. These strategies can contribute significantly to LG employees’ well-being and their ability to cope with work pressures [31,110].

In analyzing individual variables influencing the work experience of LG people, the concept of “diversity within diversity” [93] is also helpful in understanding how LG people share many of the experiences of exclusion with other minority groups [161]. Indeed, in light of the intersectional theory, it emerges that bisexual people have a unique experience [162], marked by a higher rate of post-traumatic stress disorder (PTSD) and bullying than single-sex individuals [112], as they are “doubly stigmatized”, both in heteronormative contexts and within sexual minorities themselves [99]. Furthermore, significant differences emerged between male and female subjects. Indeed, it has been found that regarding homosexual women, there is a mixture of homophobia and sexism [163], fewer opportunities to develop their careers [150], and, as with bisexual individuals, double discrimination. It has also been found that LG workers with poor mental health are mostly cis or trans women [43] and that gay men are more often job-satisfied than lesbian women, albeit less so than heterosexual men [65].

It also emerged that work and affective life are in a continuous interchange, without the balance of which well-being seems impossible. Indeed, disclosure in the work context was often positively associated with partner satisfaction but negatively with family interference with work [131]. Furthermore, although support for life beyond work is positively correlated with job satisfaction, even in the presence of high levels of support, the job satisfaction of LG employees remains lower than that of their heterosexual colleagues [52]. In the same vein, it emerged that a significant individual variable is the perception of incongruence between one’s family identity and workplace expectations, as well as that the stigmatization of one’s family is often the basis of the work–family conflict [80,120,147]. Indeed, it emerged that although outness in work contexts is also generally associated with greater life and couple satisfaction, the interference of this private dimension at work, sometimes driven by the desire not to make one’s family invisible, can have adverse effects on one’s work life, compromising life satisfaction and, thus, establishing a closed circle from which an individual is impoverished and more at risk for outcomes, such as anxiety and depression [75,113,119,164] (see Table 1).

### 3.4. Contextual Variables Influencing the Work Experience of LG People

The analysis of the literature reviewed revealed the entrenchment of homophobia, heterosexism, heteronormativity, and the valorization of traditional masculinity in specific work contexts [34,88,128]. For example, regarding the relationship between homophobic prejudice and the expression of masculinity, it emerged that, within heterosexist contexts, in which masculinity must be exhibited to consolidate one’s status, sexual harassment often becomes part of that exhibition and contributes to the creation of a homophobic climate [130]. It also emerged that gender harassment often occurs in coexistence with heterosexist harassment and that the severity of heterosexist harassment is significantly associated with high rates of job burnout and job dissatisfaction [156,165]. In addition, it was found that greater compliance with male gender norms is associated with an increase in risky behaviors, such as isolating behaviors or discriminatory organizational practices [56]. In sum, the presence of a “heteroprofessional” tendency to exclude and discriminate against homosexuality emerges in work contexts [29,82,97,166].

Furthermore, the concept of segregation, i.e., the overrepresentation of a group in some occupations and its underrepresentation in others, regarding the occupational distribution of the relevant economy among professions [68], is particularly significant in comprehending how homophobia, heterosexism, and heteronormativity can affect organizational experiences. Indeed, it emerged that the high concentration of gays and lesbians in high-independence jobs may be due to bias during the selection phase [39]. In this vein, the discourse of employment discrimination of LG persons could be extended in an ecological perspective from the purely subjective individual life and psychic suffering to the macroscopic whole definable with the dominant heteronormative culture in biological, sociological, psychological, economic, and political domains, thus indicating that even an unspoken can create and nurture a “discourse of exclusion” [95] causing the issue of “closeting” to go beyond the perception of the individual.

Thus, homonegative events significantly inhibit disclosure processes [159] with specific psychological consequences. But this is only one of the ways the heteronormative culture manages workers’ sexual identities [36]. Indeed, it has also been found that norms governing gender and sexuality within workplaces continually influence work-related migration [81] and that sexual minority status is often associated with turnover intention [37]. Furthermore, greater experiences of heterosexism were found to be associated with fear and anger [167], which, in turn, were associated with greater mental and physical distress, turnover intentions, and lower job satisfaction [118,168], thus indicating a possible mediating role of fear between heterosexism and psychological distress [107]. In this vein, even usually effective coping strategies, such as disclosure with coworkers, can, when implemented in contexts with high levels of discrimination, lose their power as a “buffering effect” [121].

The construct of “modified labeling” [125], according to which stigmatization is believed to be the product of a social process whereby those with power can negatively label those with less power as “deviant” from the dominant social norm, emerged as particularly interesting. In such a situation, the negatively labeled person experiences a stigma that links him or her to undesirable characteristics or a devalued social position. This construct very effectively brings together the concept of stigma with that of heteronormativity. In this vein, from a Foucauldian perspective, it is not only sexuality that can be the target of repression/discrimination but also non-sexuality [138]. This seems to “in-form” us about the fact that heteronormative and stigmatizing power logics within work contexts push in the direction that all employees be made sexually intelligible and bound to a clear sexual identity.

Furthermore, organizational climate and culture are significantly related to workers’ perceived subjective experiences [24,35,109]. In this vein, Holman [169] understands climate as a “general level of support or hostility”, which is a recurring finding in the reviewed studies. We could see the importance of the voice–silence issue concerning minority identities in work contexts as an expression, first and foremost, of employees’ intention to be heard on relevant issues, with trade union significance as well [146]. In this context, the concept of perceived organizational support [64,78] carries considerable weight as, first and foremost, an antecedent of countless psychological outcomes of LG employees and disclosure processes [115]. In addition, the perception of a non-discriminatory climate toward sexual minorities, represented, for example, by the use of appropriate pronouns when addressing employees, appears to be correlated with increased developmental networks and positive organizational attitudes [79,82,146,170]. Indeed, a possible intertwining of professional life and sexual identity emerges, thus suggesting that sexuality and professionalism can mutually enhance or deny each other [60].

Furthermore, the sedimentation of a culture of silence prevents LG employees from constructing a work identity that includes their sexual identity, and this, in turn, prevents the organizations themselves from being fully inclusive [45]. It is then possible to think that at the moment when an employee expresses himself or herself on highly relevant issues, such as sexual identity, the interests of the individual and those of the organization are congruent, thus creating the conditions for the individual to feel part of a context that recognizes them and in which he or she can identify, with a spillover effect on well-being and an organizational payoff in terms of productivity and corporate citizenship [57,92,102,115]. But it is also possible to see that the dyscrasia between the interest of the individual and that of the organization leads to an increase in turnover intentions, a decrease in perceived support [77], and lower job satisfaction, lower inclusiveness [134], and lower work engagement [69].

Finally, the concept of protection seems to play an important role in safeguarding the dignity of LG workers [87]. In this vein, the concept of “workplace incivility” [171] can be used to define a series of continued low-intensity acts that violate norms of respect and whose intent to harm is ambiguous (see Table 1).

## 4. Discussion

Regarding the theoretical frameworks in the approach to sexual minorities’ work-related issues, minority stress theory [19], sexual prejudice and stigma theories [123,124], queer theory and Foucauldian paradigms [132,133], social identity theories [139], and intersectionality theory [143] emerged as the most represented in the reviewed studies. Despite the preponderance of such theoretical frameworks, other models are proposed that focus on specific variables or issues, such as the management [16] or the communication [145] of sexual identity and the disclosure process [55,62].

Regarding individual and contextual variables influencing the work experience of LG people, outness and disclosure emerged as the main variables highlighted by the reviewed studies. Indeed, a general trend seems to be that the more central and salient one’s sexual identity is, the less likely one will be to disown it and, therefore, to deny it in the workplace [114]. But it has also been found that coming-out processes are often influenced by multiple individual and contextual factors that prevent their unfolding, such as being placed in an organizational context that does not guarantee safety, having previously experienced heterosexist assaults, perceiving homonegativity among colleagues, knowing that other colleagues who have come out have been discriminated against and/or dismissed, knowing that they are placed in a broader cultural context characterized by a strongly heteronormative bias [41,53,56,59,77,82,126,135]. Furthermore, the review highlighted that the work context’s non-preparedness toward these issues often motivates subjects to disclosure, which becomes a political and claiming act witnessing a necessary change [28,87,146]. In this vein, the voice–silence question emerged as a cross-cutting issue. Indeed, organizational policy-induced silence, self-induced silence, and internalization of the culture of silence were significant factors influencing the subject’s ability to practice disclosure [82,100,145,146,154]. Moreover, organizational climate—i.e., the set of perceptions individuals have of a context—also emerged as a key factor affecting the experiences of the individuals who inhabit it [30,54,172,173], inscribed in organizational culture, just as organizational culture is inscribed in the larger context in which it is situated [174,175]. Indeed, a heterosexist workplace climate has been found to mediate the relationships between outness and job satisfaction, and anticipatory discrimination [153] has been found to result in moderate relationships between disclosure and job satisfaction.

Thus emerges a highly varied conceptual framework in dealing with sexual minorities’ work-related issues, in which, despite the recurrence of cross-cutting issues, both at the individual and contextual levels, an underlying unity seems to be lacking. Moreover, in reviewing the literature on the subject, it seems to emerge that the variety of theoretical frameworks of reference are often accompanied by significant terminological and conceptual confusions in dealing with sexual minorities’ issues that complicate, if not limit, the possibility of arriving at a coherent overall conceptual framework of reference. In this sense, it would be desirable to implement a more unified theoretical model to keep within the most relevant issues that have emerged from this systematic review while paying particular attention to a more correct and consistent use of specific terminologies and concepts. Moreover, this model could be very useful in constructing new measurement instruments to evaluate LG people’s work experience and to develop interventions to minimize the effects of stigma and heterosexism in work contexts on LG employees’ experience.

### Limitations

There are some limitations in this review. The first limitation relates to the type of literature selected, which only concerns scientific articles. Indeed, future reviews on this topic could include additional sources, such as books, clinical guidelines, and training materials. Furthermore, although no filters have been settled to restrict the search by language, the use of an English string (and keywords) may have limited the access to relevant scientific articles written in languages other than English. This limitation should be taken into account when considering the generalizability of the review findings.

Moreover, although this review focused on the work experience of LG people, the participants in the studies analyzed were significantly diverse in sexual and gender identity, including, for example, bisexuals, transgender people, etc. Indeed, the way the studies describe the research populations revealed a frequent overlap in sexual orientation and gender identity, which often made it impossible to identify specific sub-populations: in the particular case of this review, this sub-population was the LG population. This within-group diversity implies caution in interpreting our results and highlights the importance, in future reviews, of paying attention to the intersection of sexual and gender minority status, as well as to sampling strategies.

## 5. Conclusions

Taken together, the results of the present review indicate that the majority of LG participants involved in the studies suffer from the dominant heteronormativity and segregation policies and have experienced discrimination and micro-aggressions in the workplace, with important and recurring effects on their psychological health (e.g., anxiety, depression), job satisfaction (e.g., sense of belonging to the organization, career expectations) and life satisfaction (e.g., perception of authenticity, work-family balance, couple relationship, etc.). This means that, despite an increase in the promotion of equal opportunities at work, there is still persistent discrimination against LG workers. Finally, the results suggest the need to develop a unified theoretical model that would serve as a solid foundation both for the construction of specific measurement tools to assess the work experience of LG people and for the implementation of interventions aimed at minimizing the effects of heterosexism, heteronormativity, and stigma in work contexts on LG employees’ experience and promoting their psychophysical health and well-being.

## Figures and Tables

**Figure 1 ijerph-21-01355-f001:**
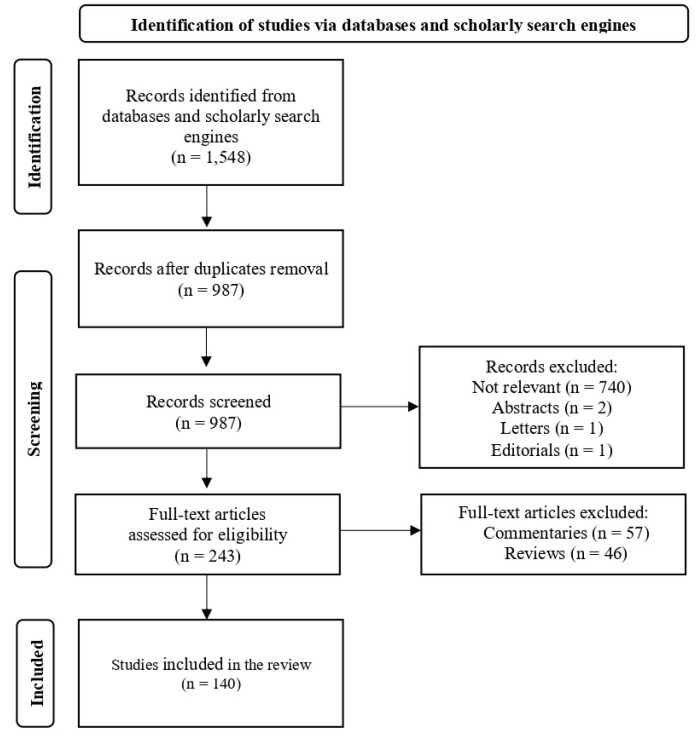
PRISMA flow diagram of study identification and selection process.

**Table 1 ijerph-21-01355-t001:** Summary of review findings: theoretical frameworks in the approach to sexual minorities’ work-related issues, individual and contextual variables influencing the work experience of LG people.

**Review Aims**		
**(1) To identify the theoretical frameworks used in the approach to sexual minorities’ work-related issues**	Theoretical Frameworks	
	Minority stress theory Sexual prejudice and stigma theories Queer theory and Foucauldian paradigm Social identity theories Intersectionality paradigm Multilevel relational framework to diversity management Communication theory of identity Social exchange theory Disclosure process model No specific theoretical framework: heterosexism, heteronormativity, cisnormativity, and pressure to conform to masculine gender norms	
**(2) To identify the main individual and contextual variables influencing the work experience of LG people.**	**Individual Variables**	**Contextual Variables**
	Outness and disclosure	Homophobia
	Authenticity	Heterosexism
	Coping strategies	Heteronormativity
	Diversity within diversity	Valorization of traditional masculinity
	Work life/affective life	Modified labeling
		Organizational climate
		Workplace incivility

## Data Availability

No new data were created or analyzed in this study. Data sharing is not applicable to this article.

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
