# Peer review of "Lesbian and Gay Population, Work Experience, and Well-Being: A Ten-Year Systematic Review"

_ijerph, 2024, doi:10.3390/ijerph21101355_

Round 1

Reviewer 1 Report

Comments and Suggestions for Authors

Dear authors

Thank you for the opportunity to review this paper, that adresses important aspects of the worklife of LGBTQ+ employees. In general, the paper is clear and well written, although especially the introduction and overall framing of the study could be improved. 

I have some comments that I believe would improve the paper.

Headline: spell out LG. Also in the abstract, the first time you use the abbreviation. 

Introduction:

Spell out lgbtq+ or use an explanation as all readers might not be familiar with the abbreviation.

Lines 52-56 are unclear: Do you mean that 21.000 Italian LGBTQ+ people have been discriminated?

Lines 57-71: This part is based on a single source from 2010. I would encourage the authors to find more recent papers.  

Lines 81-85: More references are needed here.

Lines 86-91: The job demand-job resources model could be expanded here, and it would be helpful to elaborate on the theoretical frameworks stated later in the paper – which frameworks are most widely used, and how can they be helpful in this area of research?

From line 161: It’s a bit unclear to me how you assess the theoretical frameworks in the studies. I would include this in the method section. 

Comments on the Quality of English Language

The paper could be improved by lanugage edititing.

Author Response

Dear Reviewers,

the authors wish to thank you sincerely for the careful and positive assessment of our work and for the useful suggestions given to revise our paper. 

We’ve considered all your comments and the manuscript has been carefully revised according to all of them. All the changes to the manuscript have been highlighted by tracking the new text passages in red.

A detailed point-by-point description of the implemented changes is provided below:

Reviewer 1

Dear authors

Thank you for the opportunity to review this paper, that adresses important aspects of the worklife of LGBTQ+ employees. In general, the paper is clear and well written, although especially the introduction and overall framing of the study could be improved. 

I have some comments that I believe would improve the paper.

  1. Headline: spell out LG. Also in the abstract, the first time you use the abbreviation. 

  1. Answer. Thank you for your advice. In the revised paper, we have spelled out LG in the headline, as well as in the abstract, the first time we have used the abbreviation.

  1. Introduction: Spell out lgbtq+ or use an explanation as all readers might not be familiar with the abbreviation.

  1. Answer. Following your useful suggestion, we have spelled out LGBTQ+ to support readers who might not be familiar with the abbreviation.

  1. Lines 52-56 are unclear: Do you mean that 21.000 Italian LGBTQ+ people have been discriminated?

  1. Answer. In line with your comment, the sentence in lines 52-56 has been re-written to make it clear that 21.000 represents the number of LGBTQ+ people - in current or past civil unions and residing in Italy as of January 1, 2020. Afterwards, in lines 57-72 the percentage of those having experienced discriminations, microaggressions, or disadvantages in working life has been reported.

  1. Lines 57-71: This part is based on a single source from 2010. I would encourage the authors to find more recent papers.  

  1. Answer. Thank you for raising this potential misleading point. Indeed, the reference “2010” concerned the definition of microaggressions, rather than the source of data (ISTAT and UNAR survey 2022 – line 53). Therefore, in order to avoid potential misunderstanding, we have added two more recent references on the topic of microaggression, namely:

- Smith, I.A.; Griffiths, A. (2022). Microaggressions, Everyday Discrimination, Workplace Incivilities, and Other Subtle Slights at Work: A Meta-Synthesis. Hum. Resour. Dev. Rev. 2022, 21, 275–299. https://doi.org/10.1177/15344843221098756

- Fattoracci, E.S.M.; King, D.D. The Need for Understanding and Addressing Microaggressions in the Workplace. Perspect. Psychol. Sci. 2023, 18, 738–742. https://doi.org/10.1177/17456916221133825.

  1. Lines 81-85: More references are needed here.

  1. Answer. Following your useful suggestion, further references regarding research and interventions preventing stress and discrimination in the workplace have been included in the edited manuscript, namely:

- Steffens, M.C.; Niedlich, C.; Ehrke, F. Discrimination at Work on the Basis of Sexual Orientation: Subjective Experience, Ex-perimental Evidence, and Interventions. In Sexual Orientation and Transgender Issues in Organizations, Köllen, T. Eds.; Springer: Cham, CH, 2016; pp. 367–388. https://doi.org/10.1007/978-3-319-29623-4_2

- Zambrana, R.E.; Valdez, R.B.; Pittman, C.T; Bartko, T.; Weber, L.; Parra-Medina, D. Workplace stress and discrimination effects on the physical and depressive symptoms of underrepresented minority faculty. Stress Health 2021, 37, 175–185.  https://doi.org/10.1002/smi.2983

  1. Lines 86-91: The job demand-job resources model could be expanded here, and it would be helpful to elaborate on the theoretical frameworks stated later in the paper – which frameworks are most widely used, and how can they be helpful in this area of research?

  1. Answer. Following your useful suggestion, the job demands- resources model has been reported in more detail (please, see lines 88-97).

Moreover, considering the theoretical frameworks stated later in the paper (which emerged from the systematic review), following your important advice, section 3.2 has been edited and enhanced to better explain and elaborate all the theoretical frameworks that emerged (not only the most widely used in the reviewed studies but all of them).

  1. From line 161: It’s a bit unclear to me how you assess the theoretical frameworks in the studies. I would include this in the method section. 

  1. Answer. Thank you for raising this point. In the edited manuscript, we have better clarified – in the method section – that the data extraction and synthesis were performed narratively considering the research questions settled for the purpose of the current study, namely, (1) to identify the theoretical frameworks used in the approach to sexual minorities’ work-related issues and (2) to identify the main individual and contextual variables influencing the work experience of LG people.

Considering the assessment of the frameworks, the number of studies adopting and – most importantly - clearly stating the theoretical framework was used by us to derive the most widely used frameworks (e.g., Minority Stress Theory; 16.4%, 21 out of 140 studies).  Also, considering the suggestion given by Reviewer 2, a new Table (Table 1) was included in the edited manuscript to summarize review findings – including all the theoretical frameworks that emerged from  the literature.

Thank you once again for your useful suggestion.

  1. The paper could be improved by lanugage edititing.

  1. Answer. Following your suggestion, the paper has been reviewed by a native English-speaking researcher.

Thank you once again for the careful assessment and for the important suggestions given to increase the quality of our work.

Kind Regards

Reviewer 2 Report

Comments and Suggestions for Authors

I believe that the topic addressed in the research is important and of great interest to the scientific community and to a large part of the public. I think that all the studies carried out to break down the barriers of discrimination in any of its expression should be considered of special consideration for publication, because through knowledge these barriers can be broken.

The research presented by the authors is very well founded in its theoretical introduction, I think they deal with all the variables to be taken into consideration in the study. The objectives of the research are fine, although I would add the years in which the systematic review has been carried out.

The method is also clearly developed and easy to follow if you intend to replicate the study. In this section I would include in the heading selection and exclusion criteria. In this section, the authors indicate that there is no language restriction in the search. I would reflect on this aspect, because I do believe that there is a restriction from the moment they use a specific language to carry out the searches. There are scientific journals that do not have abstracts or keywords in the language used for the search. Therefore, to say that there has been no language restriction, I am not sure, but I am sure that there has been. This aspect should also be reflected in the limitations of the study.

I consider that the results are well planned, however, a table that compiles these results would not be bad.

I believe that the discussion and conclusions are pertinent to the referenced studies and the results obtained.

As for the limitations presented I agree with them, perhaps include the one indicated above about the language after they reflect on it and review their searches.

For all these reasons, I consider that the article can be published including these small modifications. Thank you very much for having the opportunity to read this research.

Author Response

Dear Reviewers,

the authors wish to thank you sincerely for the careful and positive assessment of our work and for the useful suggestions given to revise our paper. 

We’ve considered all your comments and the manuscript has been carefully revised according to all of them. All the changes to the manuscript have been highlighted by tracking the new text passages in red.

A detailed point-by-point description of the implemented changes is provided below:

Reviewer 2

I believe that the topic addressed in the research is important and of great interest to the scientific community and to a large part of the public. I think that all the studies carried out to break down the barriers of discrimination in any of its expression should be considered of special consideration for publication, because through knowledge these barriers can be broken. 

  1. The research presented by the authors is very well founded in its theoretical introduction, I think they deal with all the variables to be taken into consideration in the study. The objectives of the research are fine, although I would add the years in which the systematic review has been carried out.

  1. Answer. Thank you for the useful advice. In the edited manuscript, the time frame in which the systematic review was conducted has been included (please, see line 103).

  1. The method is also clearly developed and easy to follow if you intend to replicate the study. In this section I would include in the heading selection and exclusion criteria.

  1. Answer. Following your suggestion, in the edited manuscript, the heading selection has been better clarified, so that it was labelled as “Selection: Inclusion and Exclusion Criteria”.

  1. In this section, the authors indicate that there is no language restriction in the search. I would reflect on this aspect, because I do believe that there is a restriction from the moment they use a specific language to carry out the searches. There are scientific journals that do not have abstracts or keywords in the language used for the search. Therefore, to say that there has been no language restriction, I am not sure, but I am sure that there has been. This aspect should also be reflected in the limitations of the study.

  1. Answer. Thank you very much for raising this extremely interesting point. Indeed, we did not set any filter to restrict the search by language (which can be done manually during a systematic review), and indeed we also found articles with title, abstract and keywords in English that did not have an English corpus (Spanish, French, German, etc.) and such articles were routinely screened. However, as interestingly noted by the reviewer, the use of an English string automatically (even if not manually) implies a filter, which creates a limitation that absolutely must be considered and which we have therefore addressed as a limitation. Thank you again for these interesting insights.

  1. I consider that the results are well planned, however, a table that compiles these results would not be bad. 

  1. Answer. Following your useful suggestion, a table that complies with the results has been provided in the edited manuscript (please, see Table 1).

  1. I believe that the discussion and conclusions are pertinent to the referenced studies and the results obtained. As for the limitations presented I agree with them, perhaps include the one indicated above about the language after they reflect on it and review their searches. 

  1. Answer. Thank you once again for underlining this point. In the limitation section, this issue has been addressed (please see lines 428-431).

  1. For all these reasons, I consider that the article can be published including these small modifications. Thank you very much for having the opportunity to read this research.

  1. Answer. The authors would like to thank you once again for your careful assessment and for the valuable suggestions given to increase the quality of our work.

Kind Regards

Round 2

Reviewer 1 Report

Comments and Suggestions for Authors

Thank you for the revised edition of the paper. I like the improvements made to the paper, and have only a few minor comments:

- In the introduction, the description of the job-demand model works well. However, it would be helpful to tie the model more precisely to the LG population, just with 1-2 sentences saying what particular demands would be relevant for this population.

- Although the theoretical frameworks are much better described in the paper, I still believe that mentioning a couple of them in the introduction would improve the readibility of the paper. Or just mentioning that there are different theoretical frameworks in this field of research.